# Effect of a Nutritional Education Intervention on the Reduction of Added Sugar Consumption in Schoolchildren in Southeastern Mexico: Community Study

**DOI:** 10.3390/foods14020179

**Published:** 2025-01-08

**Authors:** Carmen Morales-Ruán, Teresa Shamah-Levy, Danae Gabriela Valenzuela-Bravo, Rebeca Uribe-Carvajal, Corin Hernández-Palafox, María Concepción Medina-Zacarías, Ignacio Méndez Gómez-Humarán

**Affiliations:** 1Instituto Nacional de Salud Pública, Centro de Investigación en Evaluación y Encuestas, Avenida Universidad 655, Colonia Santa María Ahuacatitlán, Cuernavaca 62100, Mexico; cmruan@insp.mx (C.M.-R.); ciee23@insp.mx (D.G.V.-B.); ciee24@insp.mx (R.U.-C.); corinpalafox@gmail.com (C.H.-P.); conymedina82@gmail.com (M.C.M.-Z.); 2Centro de Investigación en Matemáticas A.C., Calzada de la Plenitud 103, Fracc. José Vasconcelos Calderón, Aguascalientes 20200, Mexico; imendez@cimat.mx

**Keywords:** nutrition education, dietary habits, childhood obesity prevention

## Abstract

Consumption of added sugars negatively affects schoolchildren’s health, making it essential to promote strategies designed to reduce their intake through educational interventions. This study aimed to evaluate the effect of a nutritional education intervention, INCAI, on the consumption of added sugars among schoolchildren in southeastern Mexico. A 9-month educational intervention was designed to promote healthy eating, physical activity, and the development of socio-emotional skills among primary school children. Information on the usual intake of foods and beverages was collected from 400 schoolchildren at the beginning and end of the intervention using a semiquantitative food frequency questionnaire. For the analysis, a generalized estimating equation (GEE) model was constructed using the Poisson distribution family to estimate the effect of the intervention. The relative incidence in the final stage showed a slight increase of 4% (*p* = 0.093) in the percentage of added sugars compared with the baseline levels in the control group. By contrast, the intervention effect, represented by the interaction between treatment and time, showed a 10% reduction in the final stage in the intervention group (*p* < 0.001). Based on these findings, the INCAI educational intervention effectively reduced added sugar consumption by 10% in the intervention group, while no significant reduction was observed in the control group. These results suggest that nutritional education programs can be a useful strategy for lowering added sugar intake among schoolchildren, highlighting the importance of incorporating such interventions into public health policies targeting child populations.

## 1. Introduction

Excessive consumption of added sugars is a growing global concern associated with a significant increase in the prevalence of chronic diseases, such as obesity, type 2 diabetes, cardiovascular diseases, and dental caries [1,2,3]. The primary source of added sugar consumption worldwide is sugar-sweetened beverages, including soft drinks and other sweetened drinks [4,5]. In addition to beverages, many processed foods, such as cakes, cookies, and cereals, contain high levels of AS. These products are often appealing to children and adolescents due to their taste and convenience [6]. According to data from the National Health and Nutrition Survey (ENSANUT 2020–2022), in Mexico, 66% of schoolchildren exceeded the recommended limit for added sugar consumption (10% of total energy); sugar-sweetened beverages were the group that contributed the most to this intake [7]. It was also documented that 41% of schoolchildren are overweight or obese [7]. In the southeastern region of the country, the prevalence of overweight and obesity among schoolchildren is 37.9% [8]. Developing strategies aimed at reducing added sugar consumption in the school population of this region could have a positive impact on their health and nutritional status, helping to prevent these chronic diseases and improving their quality of life. Nutritional education (NE) plays a crucial role in reducing added sugar intake, as it provides the knowledge and skills necessary to make informed and healthy dietary decisions [9,10]. Health interventions with an NE component have been shown to have a favorable impact on both individual and family eating behaviors, contributing to a reduction in the consumption of foods rich in sugar [10,11,12]; however, it is essential that NE interventions are adapted to the specific needs and characteristics of the population. Greater impact on the promotion of school health can be achieved by developing and implementing culturally relevant interventions tailored to local needs [13].

In 2023, the project ‘Comprehensive Community Intervention for Children in Nutritious and Healthy Environments (INCAI due to its acronym in Spanish)’ was developed and implemented to prevent overweight and obesity in schoolchildren in the state of Yucatán. This was achieved by promoting healthy environments through a balanced diet, physical activity, and the strengthening of social and emotional skills. The intervention included NE activities, such as workshops on sugar-sweetened beverages and food labeling, gardening workshops, dissemination of infographics and informational videos, creation of posters, and training for school staff and families. Therefore, although the intervention had more objectives, the analysis of this manuscript aimed to evaluate the effect of NE on the consumption of added sugars in a school population in southeastern Mexico. Our study hypothesis is that an intervention with a NE component has a positive effect on reducing the consumption of added sugars in the target population.

## 2. Materials and Methods

A community trial was conducted in 12 public primary schools in the state of Yucatán, located geographically in southeastern Mexico. Schools were selected based on the willingness of school authorities to participate in the study. Six schools were randomly assigned to the intervention group and six to the control group. The study population comprised students in the 4th and 5th grades of primary school.

This study was conducted in three stages. In the first stage, a systematic literature review was carried out along with a participatory community diagnosis in six primary schools with characteristics similar to those selected for the study. In the second stage, a baseline measurement was conducted in the 12 participating schools, and in the six intervention schools, the activities designed to promote behavioral changes in students were implemented and monitored, focusing on physical activity, healthy eating, and socio-emotional skills. Finally, in the third stage, a final measurement of the indicators and variables of interest was conducted for students of the 12 participating schools.

At each stage, staff (with nutrition and psychology backgrounds) were trained according to specific requirements, either in the application of questionnaires, standardization of measurements, or development of intervention activities.

The project was carried out with the collaboration of institutions responsible for education and health and the University of the State of Yucatán, which participated in the development of the intervention, dissemination of the study among the school community, providing facilities for the research staff, and managing the infrastructure of the schools.

### 2.1. Sample Design

The sample size was calculated for a larger intervention considering a minimum difference of 9% in the prevalence of overweight and obesity between the comparison groups, with a 95% confidence level, 80% statistical power, and 5% non-response rate. This sample did not consider the percentage change in free sugar intake. As a result, the sample size of 488 students was estimated to be equally distributed, with 244 in the intervention group and 244 in the control group.

### 2.2. Description of the Intervention

The intervention was carried out during the 2023–2024 school year, lasting 9 months (September 2023 to May 2024), within school facilities, and during school hours. The design of the implementation activities was based on the issues identified in the community diagnosis and incorporated elements from the methodological frameworks of intervention mapping [14] and the socio-ecological model [15]. The designed activities focused on three main components: socio-emotional skills, physical activity, and nutrition (Table 1).

The objective of the socio-emotional skills component was to promote the recognition and regulation of emotions. Regarding physical activity, the aim was to encourage the daily recommendation of 60 min of moderate to vigorous physical activity and to limit screen time to less than 2 h per day. Lastly, in the nutrition component, the promotion of water consumption, as well as fruit and vegetable intake, was emphasized while discouraging the consumption of sugar-sweetened beverages and ultra-processed products.

Various dissemination and NE strategies have been implemented to achieve the nutrition component’s goal, targeting students, parents, and teachers. In schools, infographics on the importance of drinking water and healthy eating were displayed in visible locations. Additionally, two “For Your Health” fairs were organized, with playful activities focused on school and family nutrition, allowing the active participation of the school community.

Parents received a 20 s informational video weekly via WhatsApp, covering topics related to each component (physical activity, socio-emotional skills, and nutrition). For the nutrition component, four educational shorts titled “Let’s Add Color to Our Snack” were sent, focusing on including a variety of fruits and vegetables in snacks, as well as the benefits these foods provide to the body. The other four educational shorts, under the title “Healthy Eating,” provided information on how to achieve healthy eating, such as identifying ultra-processed products and avoiding their consumption, the importance of proper water intake, and the relationship between diet and caring for the planet.

Five nutrition workshops were held: four aimed at students and teachers and one at parents. These workshops covered topics such as the sugar content in beverages and ultra-processed products, the use of front-of-pack labeling to facilitate informed choices, the differences between natural and ultra-processed foods, and the nutritional contributions of each. The steps for setting up school gardens and their benefits, the importance of drinking water, and the significance of healthy and sustainable breakfasts and snacks for improving health and protecting the planet were also reviewed. Each workshop was designed to provide practical and accessible information. The educational resources included PowerPoint presentations, games, group dynamics, printed materials, and assessment tools.

Additionally, a talk was held with school principals and school store managers on the guidelines regulating the foods and beverages that can be offered within the school, emphasizing the importance of providing nutritious foods, such as vegetables, fruits, and water.

### 2.3. Variable Description

#### 2.3.1. Diet

To evaluate the effect of the intervention on added sugar consumption, dietary information was obtained through the semiquantitative Food Frequency Questionnaire (FFQ) covering the seven days prior to the interview, which was administered to parents. This instrument is used in the National Health and Nutrition Survey [16] and includes a closed list of foods and beverages to which some commonly consumed foods in the state of Yucatán were added. The questionnaire included questions about the days and times per day that the foods were consumed, as well as the size and number of servings consumed each time. Using this information, consumption in grams was estimated, which was later converted to energy (kcal/day) and sugar (g/day) using a composition database developed by the National Institute of Public Health. The average consumption of each food and beverage was estimated, and those with an intake of > 4 standard deviations (S.D.) were identified and imputed with the mean. Questionnaires with more than seven imputed foods were excluded as invalid.

#### 2.3.2. Percentage of Added Sugars

The amount of added sugar in the processed foods was obtained by assigning a percentage of the total sugar declared by the manufacturer. According to the methodology recommended by the Pan American Health Organization (PAHO) [17], these percentages range from 0% added sugars in natural foods (such as fruits) to 100% in products where the sugar does not originate from natural sources (such as candies and industrialized drinks).

Foods and beverages were classified into (1) sweetened cereals (breakfast cereals, sweet bread, donuts, bakery churros, industrialized pastries and donuts, cookies, and cereal bars); (2) snacks, sweets, and desserts (e.g., yogurt-based desserts, popcorn, canned fruits, chocolates, chips, marshmallow-based sweets, gelatin, flan, ice cream, sorbet, milk popsicles, and fruit popsicles); and (3) sugar-sweetened beverages (e.g., atole, sugary sodas, sweetened fruit juices, flavored water, sugary dairy beverages, sweetened coffee, and tea), considered as the main sources of added sugars. This classification was adapted from a previously published framework that categorized foods and beverages into 13 groups based on their nutritional characteristics and relevance to health outcomes [18]. Based on this classification, the total and added sugar intakes of each group were estimated. Finally, the added sugar consumption in grams and its percentage relative to total energy intake were calculated.

#### 2.3.3. Self-Efficacy in Diet

Self-efficacy refers to the ability and confidence of an individual to perform a specific behavior [19]. In the context of preventive health behaviors, individuals with high perceived self-efficacy are more likely to initiate preventive care, seek early treatment, and be more optimistic about the effectiveness of these measures [20]. In this regard, to evaluate the self-efficacy of schoolchildren regarding diet, an ad hoc questionnaire was designed, taking reference to questions used in similar interventions [21]. The questionnaire included eight items, with statements regarding how capable the student feels of maintaining healthy eating in different situations. Each item had a dichotomous response option (0 = no, 1 = yes). For the analysis, the results were calculated by summing the positive responses, where the minimum and maximum values obtained were zero and eight.

#### 2.3.4. Sociodemographic Variables

Age, sex, and indigenism data were collected from the participating students.

### 2.4. Ethical Considerations

This study was conducted in accordance with the Declaration of Helsinki, and the protocol was approved by the Ethics, Biosafety, and Research Committees of the INSP (Project CI: 1791) on 6 June 2022. Prior to data collection, informed consent was obtained from the parents, and informed assent was obtained from the students to participate in the study.

### 2.5. Statistical Analysis

To estimate the intervention effect, a population-averaged model (generalized estimating equations, GEE) was constructed using the Poisson distribution family, log-link function, and an exchangeable correlation structure. The time–intervention interaction was used as an estimator of the average effect on the percentage of added sugar per day, adjusted for total energy, self-efficacy, indigenism, sex, and age. SPSS version 25 and Stata version 14.0 (with the SVY module) statistical software packages were used for the analysis.

## 3. Results

At baseline, data were collected from 494 schoolchildren, with 250 in the intervention group and 244 in the control group. In the final measurement, a response rate of 87.4% was achieved, allowing data collection from 432 schoolchildren (212 in the intervention group and 220 in the control group). For this analysis, 32 schoolchildren were excluded because of implausible dietary data. The final sample consisted of 400 schoolchildren. (Figure 1).

A comparative analysis was conducted to evaluate if those subjects who were lost to follow-up had important differences in anthropometric and demographic characteristics from the subjects that completed the study, but no differences were found.

The average age of the schoolchildren at the beginning of the study was 10 years in both groups (intervention and control). Regarding sex distribution, 52% of the participants in the intervention group were male, compared to 46% in the control group. A total of 30% of the population in the intervention group reported speaking an indigenous language, compared to 13% in the control group. As for school grade, at baseline, 49% of the schoolchildren in the intervention group were in fourth grade, whereas in the control group, this figure was 53%. No statistically significant differences were observed in the demographic characteristics between the groups (Table 2).

Regarding the consumption of added sugars by food group, Table 3 shows that sugar-sweetened beverages were the most consumed by schoolchildren both at baseline and at the final measurement in both groups. However, in the intervention group, the median consumption of sugar-sweetened beverages decreased from 38.0 g at the beginning of the study to 30.6 g at the final measurement, while in the control group, it went from 37.5 g to 34.6 g. In the intervention group, the proportion of energy from added sugars decreased from 14.2% to 12.6% of the total daily energy consumed, whereas in the control group, the percentage remained unchanged.

In the statistical model (Table 4), the intervention effect (Figure 2), represented by the interaction between treatment and time, showed a reduction of 10% at the final stage in the intervention group (*p* < 0.001). The relative incidence at the final stage in the control group showed a non-significant of 4% (*p* = 0.093) in the percentage of added sugars compared to the baseline. There were no significant differences in the percentage of added sugars between the intervention and control groups at the beginning of the study (*p* = 0.591).

Consumption of 1000 kcal was associated with a slight 2% increase in the percentage of added sugars (*p* = 0.09), whereas higher dietary self-efficacy was associated with a 2% reduction (*p* = 0.008). Children who did not speak an indigenous language had an 18% higher percentage of added sugars than those who did (*p* < 0.001). Females showed a 10% increase compared to males (*p* < 0.001), and for each additional year of age, there was a 3% reduction in the percentage of added sugars per day (*p* = 0.011).

## 4. Discussion

The main findings of this study show that the INCAI educational intervention contributed to a significant 10% reduction in added sugar consumption among schoolchildren in the intervention group compared to the stability observed in the control group. This is particularly relevant when considering that in the Mexican school population, it has been documented that 60% exceeds the recommended limit of added sugar intake of 10% of energy, with sugar-sweetened beverages being the group that contributes the most to this excessive consumption [7].

Several interventions conducted internationally have demonstrated the effectiveness of educational programs in reducing unhealthy product consumption, aligning with the findings of our study. In Chile—a Latin American country that shares several cultural aspects with Mexico—a NE intervention was carried out with preschool and school-aged populations. This study showed a decrease in the consumption of sugar-sweetened beverages (*p* < 0.05) and unhealthy foods such as sausages (*p* < 0.005), cakes and ice cream (*p* < 0.005), salty (0.0002) and sweet snacks (*p* < 0.005) in the intervention group, compared to the control group [22].

Similarly, other countries have successfully implemented large-scale interventions. In the United States, the nationwide “Let’s Move!” campaign has also succeeded in reducing added sugar consumption in schools through educational strategies, including promoting physical activity and improving the nutritional quality of school meals as access to water during meals and throughout the day, and do not serve sugary drinks. For children aged two and older, this includes serving low-fat (1%) or non-fat milk and no more than one 4–6 ounce serving of 100% juice per day [23].

In the United Kingdom, the “Change4Life” program also showed a significant reduction in the purchase of foods and beverages high in added sugars; in the intervention group, 32% purchased a lower-sugar drink, compared to 19% in the comparison group (*p* = 0.01), and 24% switched to a lower-sugar cereal, compared to 12% in the comparison group (*p* = 0.009) [24], aligning with the results observed in the present study.

In Austria, the ACTION NE program for fifth-grade children demonstrated a significant decrease in free sugar intake in the intervention group over time, with a group difference of −10.1 (95% CI: −18.8, −1.5; *p* = 0.021) g/day. This reduction was primarily attributed to a decrease in the consumption of “sweets and pastries”, “sodas”, “fast food”, and “salty snacks” [25].

Additionally, a school-based intervention in China that involved parents observed a decrease in the frequent consumption of sugar-sweetened beverages (≥4 times per week) in schoolchildren in the intervention group compared to the control group (31.5% vs. 56.2%, *p* <0.01) [26].

These examples emphasize that well-structured educational interventions, adapted to cultural and local contexts, can significantly improve children’s dietary habits.

Additionally, we found that schoolchildren who do not speak an indigenous language have a higher percentage of added sugar consumption than those who speak an indigenous language. This suggests that cultural and socioeconomic differences may strongly influence dietary habits, highlighting the importance of tailoring interventions to address these factors. Research has documented variations in the consumption and perceptions of sugar-sweetened beverages across racial, ethnic, and socioeconomic groups [27].

The higher added sugar consumption observed in girls than in boys indicates that educational strategies should also consider gender differences in dietary behaviors. Evidence shows that boys generally consume more sugar-sweetened beverages than girls, as supported by the data in other studies [28,29]. However, the overall effect of the intervention appears to be equally beneficial for both sexes, which is positive from the perspective of health equity.

Our study has some limitations that should be considered. For example, while we successfully measured consumption patterns, we did not directly assess the changes in knowledge, as initially suggested. Additionally, the non-significant increase in added sugar consumption in the control group does not necessarily indicate a loss of effect but may reflect external influences beyond the study period. These limitations suggest the need for continuous reinforcement to maintain long-term dietary changes. Another limitation is that our analysis was conducted using only data from completers rather than employing an intent-to-treat approach, which might have impacted the generalizability and robustness of the findings.

One of the main strengths of this study is the comprehensive approach of the INCAI intervention, which not only provides information on healthy eating but also incorporates physical activity and the development of socio-emotional skills. This multidisciplinary approach aligns with evidence showing that combining multiple health-promoting components enhances the overall impact of interventions. Additionally, the methodological design, which included a longitudinal follow-up over nine months and the use of advanced analyses, such as generalized estimating equations (GEE), ensured a robust analysis of the results, increasing confidence in the validity of the findings.

Another significant strength is the implementation of the program in a school setting, an environment that offers the possibility of directly impacting a substantial population of children and ensuring a broader reach of the intervention. Furthermore, the cultural adaptability of the INCAI program makes it a valuable model for addressing dietary disparities among specific subgroups, such as indigenous children and girls. The sustainability of such programs in schools is essential for ensuring long-term changes in dietary habits, and the success of the INCAI reinforces the feasibility of continuing to implement similar programs.

The results of this study have important implications for public health, particularly in promoting healthy eating habits from an early age. Educational interventions such as those included in INCAI have proven to be effective tools for reducing added sugar consumption, which can significantly impact the prevention of diet-related chronic diseases such as obesity and type 2 diabetes. Moreover, integrating these interventions into public health policies can promote healthier school environments, combining dietary improvements with physical activity and socio-emotional development.

This suggests that comprehensive interventions, particularly those targeting cultural and gender-specific factors, can significantly enhance their effectiveness and equity.

## 5. Conclusions

In conclusion, the INCAI educational intervention was effective in reducing added sugar consumption among schoolchildren in southeastern Mexico, achieving the main objective of this study. This intervention should be considered part of a comprehensive approach to improve child health and prevent long-term chronic diseases. Furthermore, the cultural and gender adaptability of future interventions will be crucial for ensuring that all school populations benefit equally. The continued implementation of programs such as INCAI will contribute to the creation of healthier environments and the development of sustainable dietary habits in childhood.

## Figures and Tables

**Figure 1 foods-14-00179-f001:**
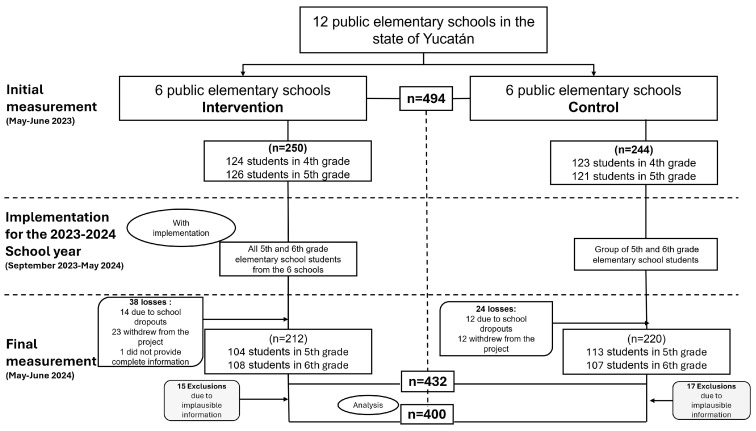
Study population selection throughout the research.

**Figure 2 foods-14-00179-f002:**
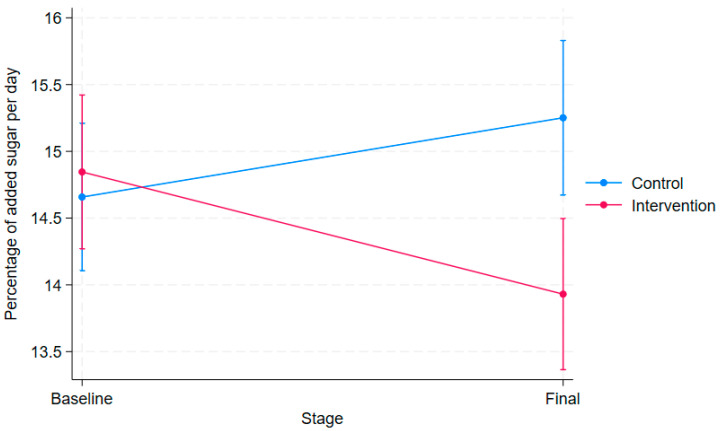
Percentage of added sugar consumption per day in the intervention and control groups.

**Table 1 foods-14-00179-t001:** Components and activities of the community intervention for comprehensive action in nutritious and healthy environments for children (INCAI).

Components of the Intervention	Component Objective	Activities
Socio-emotional skills (5 h)	Promote the recognition and regulation of emotions	Activities for schoolchildren: Presentation and dissemination of the infographic “Let’s Talk About Your Emotions”; health fair: “Emotional Recognition Station”; active breathing; and the game “Emotion Basket.” Activities for parents: Presentation of the “Emotion Guide”.
Physical activity (7 h)	Promote adherence to daily recommendations for physical activity (60 min per day of moderate to vigorous physical activity) and screen time (less than 2 h per day)	Activities for schoolchildren: Presentation and dissemination of the infographic” Get Active 60 Minutes a Day”; health fair: physical activity and sedentary behavior promotion Station; physical activation at the beginning of the school day; active breaks throughout the school day; active recess and painting playground games at school. Activities for parents: Physical activity and sedentary behavior workshop; sending informational educational shorts “Let’s Play!” Activities for primary school teachers and principals: Awareness-raising/training for the implementation of physical activation, active breaks, and active recess.
Nutrition. (11 h)	Promote the consumption of plain water and discourage the intake of sugary beverages among schoolchildren, as well as encourage the consumption of fruits and vegetables as part of school snacks and discourage the consumption of ultra-processed products	Activities for schoolchildren and primary school teachers: Presentation and dissemination of the infographic “Why is Eating Healthy Important?”; presentation and dissemination of the infographic “Why is Drinking Plain Water Important?”; health fair: healthy eating station; workshop on gardens: “Seed Planting”; workshop “Sugary Beverages and Food Labeling”; workshop on natural vs. processed foods; talk on the importance of water. Activities for parents: Talk on “Healthy Breakfast and Snack” for parents; sending informational educational shorts “Let’s Add Color to Our Snack” for parents; sending informational educational shorts “Eating Natural is Eating Healthy.” Activities for principals and those in charge of the school cooperative and/or store: Talk on the general guidelines for the sale and distribution of food and beverages.

**Table 2 foods-14-00179-t002:** Sociodemographic characteristics of the population by treatment group and measurement.

	Intervention Group			Control Group			
	n = 197			n = 203			
	Mean		S.D.	Mean		S.D.	
Age	9.96		0.82	10.0		0.77	
	n	%	(95% CI)	n	%	(95% CI)	*p* Value
Sex							
Male	103	52.3	(41.9, 62.5)	94	46.3	(41.7, 51)	0.27
Female	94	47.7	(37.5, 58.1)	109	53.7	(49, 58.3)	
Indigenism							
Yes	59	29.9	(7.1, 70.5)	27	13.3	(3.1, 42.6)	0.34
No	138	70.1	(29.5, 92.9)	176	86.7	(57.4, 96.9)	
School grade							
4°	96	48.7	(46.5, 51)	107	52.7	(47.9, 57.5)	0.127
5°	101	51.3	(49, 53.5)	96	47.3	(42.5, 52.1)	

**Table 3 foods-14-00179-t003:** Description of added sugar consumption by food group in schoolchildren.

	Baseline Assessment						Final Assessments					
Variable	Intervention			Control			Intervention			Control		
	n = 197			n = 203			n = 197			n = 203		
	Median	p 25	p 75	Median	p 25	p 75	Median	p 25	p 75	Median	p 25	p 75
Added sugars intake by food group (g/day)												
Sweet cereal	6.7	3.0	12.1	6.1	2.5	13.3	5.1	2.6	10.2	6.3	2.4	11.8
Snacks, candies, and desserts	13.1	6.6	20.4	11.7	4.1	19.9	9.2	2.5	17.2	9.3	4.2	21.6
Sugar-Sweetened beverages	38.0	26.7	56.7	37.5	23.0	67.3	30.6	17.5	45.5	34.6	21.4	53.9
Total added sugar intake												
Grams/day	68.8	48.4	95.7	63.6	43.3	96.4	52.5	32.5	72.0	59.8	40.9	87.8
Kcal/day	275.2	193.6	382.8	254.3	173.1	385.5	209.9	130.1	288.1	239.3	163.6	351.3
Percentage of total energy per day	14.2	10.9	18.3	14.5	10.4	18.8	12.6	8.6	17.0	14.3	10.5	18.6

**Table 4 foods-14-00179-t004:** Relative rates estimated from the population-averaged model.

% Added Sugars	Relative Rate *	*p*	95% CI	
Final (vs. baseline)	1.04	0.093	0.99	1.09
Intervention (vs. control)	1.01	0.591	0.96	1.07
Final*Intervention (effect)	0.90	0.000	0.85	0.95
Total energy (1000 kcal)	1.02	0.092	1.00	1.05
Self-efficacy	0.98	0.008	0.97	1.00
Indigenism (no)	1.18	0.000	1.12	1.24
Sex (female)	1.09	0.000	1.05	1.14
Age (years)	0.97	0.011	0.94	0.99
Constant	18.97	0.000	14.09	25.55

* Relative rates represent multiplicative changes in the percent of sugar consumed.

## Data Availability

The original contributions presented in this study are included in the article; further inquiries can be directed to the corresponding author.

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
