# Peer review of "Effect of a Nutritional Education Intervention on the Reduction of Added Sugar Consumption in Schoolchildren in Southeastern Mexico: Community Study"

_foods, 2025, doi:10.3390/foods14020179_

Round 1

Reviewer 1 Report

Comments and Suggestions for Authors

Abstract:

While I understand the desire report an increase in the control group, this was not significant. I don’t think there is anything wrong with saying there was no change in the control group.

Also if you report the p value for the control group, why not the intervention group. Was there significance?

Introduction:

Line 28: You define added sugars in this line, but then proceed throughout the manuscript to not use it. Suggest removing it or changing all reference to added sugar as AS.

Line 58: Your intervention is quite comprehensive and yet, you state that the aim was AS reduction? Was this the only aim or is this the aim that you reported in this manuscript? I think clarity is needed here.

Methods:

Line 153: Can you provide additional information regarding the classification? What is this based on? How did you determine this classification? Some cereals would be considered desserts.

Line 163: Self efficacy of what in the diet exactly? This needs to be better defined to see how it might impact your results.

Results:

Line 215: Increase? Really no change given the p-value.

Line 217: Would suggest moving the sentence “There were…” to be the first sentence, then include what happened in the control group, intervention group.

Line 220: Remove – before %; you already say it is a reduction. Also, include this p value in abstract.

Table 3 and Figure 2 not mentioned in text. Please include.

Discussion:

Line 242: Do not need to define NE here again.

Line 250: NE should be used here. Please review discussion for this – I am finding many errors and will not identify them all.

Line 240 – 252: This paragraph needs some work. You are just listing interventions. These need to be better integrated. For example, several interventions in US, UK, etc have shown to …… Then the Chile study as it better reflects your culture and in line with your findings.

Line 270: I would move this sentence to before the implications (This suggests sentence)

Line 275: Why refer to men? Your population is children and there is data out there to support boys consuming more sweetened beverages than girls. This ref (24) is not appropriate.

Line 280: How did knowledge decrease? You did not measure this. You measured consumption.

Line 281: It was not an increase. And I don’t believe you can assume a loss of effect from this. The only way to determine this is if you followed your participants after the intervention. Please remove.

Line 291: How did these other factors influence diet? Why not include them in your model then? Did those with higher PA for example, have different eating habits? Could this have impacted your study findings?

References need to be reviewed. Some are missing page numbers, some have ranges and others not. Formatting is inconsistent. Some journals are capitalized, others not. Many errors identified. 

Reviewer 2 Report

Comments and Suggestions for Authors

·      In the title the design of study should be added

·      The abstract is too short (<200 words). It can benefit with more results description

·      Keywords should differ from those, which appear in the title.

·      In the introduction section after the aim, a hypothesis statement should be added. Moreover this section should be more referenced.

·      A major issue, that I want that authors clarify, is the management of the sample. It seems that authors conducted their analysis only on completers, and it is not clear if a subgroup of students dropped the intervention (as usually is supposed to be). I would like to know how authors managed this data? And if this is the case authors are requested to conduct an intent-to-treat analysis

·      I suggest that authors add a small table that briefly described the key points of their interventions.   

Round 2

Reviewer 2 Report

Comments and Suggestions for Authors

I am thankful to authors for revising the results according to my comment. 

However regarding the data management of only completers, rather than conducting an intent-to-treat analysis should be mentioned clearly as a limitation in the discussion section. 

Author Response

Comment: However regarding the data management of only completers, rather than conducting an intent-to-treat analysis should be mentioned clearly as a limitation in the discussion section. 

Respond: Thank you we have included as a limitation